# Neural Interaction Transparency (NIT): Disentangling Learned Interactions for Improved Interpretability

**Michael Tsang[1], Hanpeng Liu[1], Sanjay Purushotham[1], Pavankumar Murali[2], and Yan Liu[1]**

[1]University of Southern California     [2]IBM T.J. Watson Research Center

{tsangm,hanpengl,spurusho,yanliu.cs}@usc.edu, pavanm@us.ibm.com

## Abstract

Neural networks are known to model statistical interactions, but they entangle the interactions at intermediate hidden layers for shared representation learning. We propose a framework, Neural Interaction Transparency (NIT), that disentangles the shared learning across different interactions to obtain their intrinsic lower-order and interpretable structure. This is done through a novel regularizer that directly penalizes interaction order. We show that disentangling interactions reduces a feedforward neural network to a generalized additive model with interactions, which can lead to transparent models that perform comparably to the state-of-the-art models. NIT is also flexible and efficient; it can learn generalized additive models with maximum $K$-order interactions by training only $O(1)$ models.

## 1 Introduction

Feedforward neural networks are typically viewed as powerful predictive models possessing the ability to universally approximate any function [13]. Because neural networks are increasingly used in critical domains, including healthcare and finance [2, 22, 9, 5, 30], there is a strong desire to understand how they make predictions. One of the key issues preventing neural networks from being visualizable and understandable is that they assume the variable relationships in data are extremely high-dimensional and complex [23]. Specifically, each hidden neuron takes as input all nodes from the previous layer and creates a high-order interaction between these nodes.

A fundamental challenge facing the interpretability of neural networks is the entangling of feature interactions within the networks. An interaction is entangled in a neural network if any hidden unit learns an interaction out of separate true feature interactions. For example, suppose that a feedforward neural network is trained on data consisting of two multiplicative pairwise interactions, $x_1 x_2$ and $x_3 x_4$. The neural network entangles the interactions if any hidden unit learns an interaction between all four variables $\{1, 2, 3, 4\}$. Figure 1 shows how this interaction entangling is generally true. Although a relatively small percentage of hidden units in the first hidden layer entangle the pairwise interactions, as soon as the second hidden layer, nearly all hidden units entangle the interactions when at least one of them is present. Previous works have studied how to disentangle factors of variation in neural networks [4, 18, 14], but none of them have addressed the entangling of interactions. It remains an open problem of how to learn an interpretable neural network by disentangling feature interactions.

In this work, we propose the Neural Interaction Transparency (NIT) framework to learn interpretable feedforward neural networks. NIT learns to separate feature interactions within the neural network by novel use of regularization. The framework corresponds to reducing a feedforward neural network into a generalized additive model with interactions [20, 2], which can lead to an accurate and transparent model [30, 29]. By disentangling interactions, we are able to learn this exact model faster, and it is

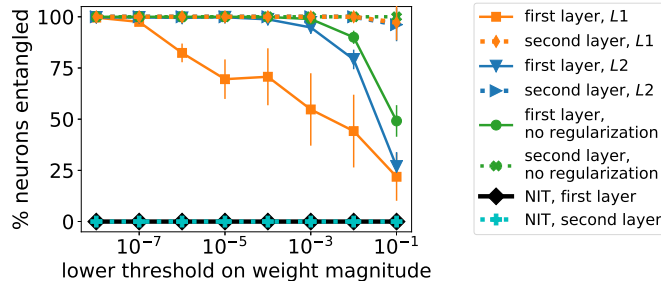

Figure 1: A demonstration of interaction entangling in a feedforward neural network. When trained on a simple dataset generated by $x_1x_2 + x_3x_4$ under common forms of regularization, the neural network tends to keep the interactions separated in the first hidden layer but entangles them in the second layer. In contrast, NIT (black and cyan) is able to fully disentangle the interactions. The meaning of entangling is detailed in §3.

visualizable. Our framework is flexible to disentangle interactions up to a user-specified maximum order. Our contributions are as follows: 1) we demonstrate that feedforward neural networks entangle interactions under sparsity regularization, 2) we develop a novel architecture and regularization framework, NIT, for disentangling the interactions to be order $K$ at maximum, and 3) we construct a generalized additive model with interactions by training only $O(1)$ models.

## 2 Related Works and Notations

In this section, we briefly discuss related literature on interpretability and disentanglement in neural networks, and generalized additive models.

**Interpretability**: Various methods [7, 11, 23, 27, 3, 31, 29] exist to interpret feedforward neural networks; however, none so far has attempted to learn or uncover a generalized additive model with interactions as a subset of a feedforward network with fully-connected layers. The closest work to our proposed framework is recent research on detecting statistical interactions from the weights of a feedforward network [31]. Because this research only extracted interactions created at the first hidden layer, it is unknown how interactions propagate through the intermediate hidden layers of the network. Other approaches to interpreting feedforward neural networks include extracting per-sample feature importance via inputs gradients [11] and global feature importance via network weights [7]. Interpretations can also be obtained by fitting simpler interpretable models [27, 3, 29] to neural network predictions, models of which include decision trees [3] and GA$^2$Ms [29]. Lastly, feedforward neural networks have been used to learn generalized additive models before [25, 31], however their construction either did not learn interactions [25] or were computationally expensive [31].

**Disentanglement**: The topic of disentanglement in neural networks comes in varied forms, but it is often studied to extract an interpretable representation from neural networks. For example, research on disentangling factors of variation has focused on modifying the training of deep networks to identify interpretable latent codes [4, 18, 14, 12] and developing ways to interpret intrinsically disentangled representations after training [26, 1]. While our work studies a different objective to achieve disentanglement, it explores the same two approaches to reach our objective.

**Generalized Additive Models (GAMs)**: GAMs are typically viewed as powerful interpretable models in machine learning and statistics [10]. The standard form of this model only constructs univariate functions of features, namely $g(E[y]) = \sum f_i(x_i)$, where $g$ is a link function and $f_i$ can be any arbitrary nonlinear function. This model can easily be visualized because each $f_i$ can be independently plotted by varying each function's input and observing the response on the output. Previous research [20] identified that GAMs in their standard form are limited in their expressive power by not learning interactions between features inherent in data. As a result, a class of models, GA$^2$M, was introduced in [20] to include pairwise interaction complexity into the GAM, in the form of $g(E[y]) = \sum f_i(x_i) + \sum f_{ij}(x_i, x_j)$. The addition of pairwise interactions was of special interest because they are still visualizable, but now in the form of heatmaps [2].

Unlike previous works which investigated disentanglement in neural networks for interpretability, our work studies disentangling interactions in neural networks to uncover GAMs.

## 2.1 Notations

Vectors are represented by boldface lowercase letters, such as $\mathbf{x}, \mathbf{w}$; matrices are represented by boldface capital letters, such as $\mathbf{W}$. The $i$-th entry of a vector $\mathbf{w}$ is denoted by $w_i$, and element $(i, j)$ of a matrix $\mathbf{W}$ is denoted by $W_{ij}$.

Let $p$ be the number of features and $N$ be the number of samples in a dataset. An *interaction*, $\mathcal{I}$, is a subset of all input features: $\mathcal{I} \subseteq \{1, 2, \ldots, p\}$, where the *interaction order* $|\mathcal{I}|$ is always greater than or equal to 2. A *univariate variable*, $u$, is a single feature that does not interact with other variables. For a vector $\mathbf{x} \in \mathbb{R}^p$, let $\mathbf{x}_{\mathcal{I}} \in \mathbb{R}^{|\mathcal{I}|}$ be the vector restricted to the dimensions specified by $\mathcal{I}$, and similarly let $x_u$ be the scalar restricted to the dimension specified by $u$.

We let a feedforward neural network and model learned by our proposed method (denoted by NIT) have the same notations. Consider a feedforward neural network $f(\cdot)$ with $L$ hidden layers and the parameters: $L + 1$ weight matrices $\mathbf{W}^{(\ell)} \in \mathbb{R}^{p_\ell \times p_{\ell-1}}$ and $L + 1$ bias vectors $\mathbf{b}^{(\ell)} \in \mathbb{R}^{p_\ell}$, $\ell = 1, 2, \ldots, L + 1$. Let $p_\ell$ be the number of hidden units in the $\ell$-th layer. We treat the input features as the 0-th layer and $p_0 = p$ as the number of input features, and we treat the output as the $(L + 1)$-th layer with $p_{L+1} = 1$. The activation function used, $\varphi(\cdot)$, is ReLU nonlinearity. Then the hidden units $\mathbf{h}^{(\ell)}$ of the neural network and the output $y$ with input $\mathbf{x} \in \mathbb{R}^p$ can be expressed as:

$$\mathbf{h}^{(0)} = \mathbf{x}, \quad y = \mathbf{w}^{(L+1)} \mathbf{h}^{(L)} + b^{(L+1)}, \quad \mathbf{h}^{(\ell)} = \varphi\left(\mathbf{W}^{(\ell)} \mathbf{h}^{(\ell-1)} + \mathbf{b}^{(\ell)}\right), \quad \forall \ell = 1, 2, \ldots, L.$$

Please note that an "interaction" in this paper is a statistical interaction that describes a non-additive influence [28] between features on a single outcome or prediction variable.

## 3 Interaction Entangling in Neural Networks

We define a neural network entangling interactions in the following way:

**Definition 1** (Entangled Interactions). *Let $\mathcal{S}$ be the set of all true feature interactions $\{\mathcal{I}_i\}_{i=1}^{|\mathcal{S}|}$ in a dataset, and let $h$ be any hidden unit in a feedforward neural network. Let $h$ capture a feature interaction $\hat{\mathcal{I}}$ if and only if there exists nonzero weighted paths between $h$ and each interacting feature and between $h$ and the output $y$. If $h$ captures $\hat{\mathcal{I}}$ such that for any two different interactions $\mathcal{I}_i, \mathcal{I}_j \in \mathcal{S}$ the following is true $\mathcal{I}_i \subsetneq \hat{\mathcal{I}}$ and $\mathcal{I}_j \subsetneq \hat{\mathcal{I}}$, then the hidden unit entangles interactions, and correspondingly the neural network entangles interactions.*

For example, suppose we have a dataset $\mathcal{D}$ that contains $N$ samples and 4 features. Each label is generated by the function $y = x_1 x_2 + x_3 x_4$ where each feature $x_i$, $i = 1, \ldots, 4$ is i.i.d. with uniform distribution between $-1$ and $1$. As a result, $\mathcal{D}$ contains two pairwise interactions: $\{1, 2\}, \{3, 4\}$. Consider a neural network $f(\cdot)$, which is trained on the dataset $\mathcal{D}$ to predict the label. If any hidden unit learns the interaction $\{1, 2, 3, 4\}$, then it has entangled interactions.

We desire a feedforward neural network that does not entangle interactions at any hidden layer so that each interaction is separated in additive form. Specifically, we aim to learn a function in the form:

$$\tilde{f}(\mathbf{x}) = \sum_{i=1}^{R} r_i(\mathbf{x}_{\mathcal{I}}) + \sum_{i=1}^{S} s_i(x_u), \tag{1}$$

where $\{\mathcal{I}_i\}_{i=1}^{R}$ is a set of interactions, $\{u_i\}_{i=1}^{S}$ is a set of univariate variables, and $r(\cdot)$ and $s(\cdot)$ can be any arbitrary functions of respective inputs. For example, we would like our previous $f(\cdot)$ to be decomposed into an addition of two functions, e.g. $r_1(\{1, 2\}) + r_2(\{3, 4\})$, where both $r_1, r_2$ perform multiplication. A model that learns the additive function in Eq. 1 is a generalized additive model with interactions.

A recent work [31] has shown that the weights to the first hidden layer are exceptional at detecting interactions in data using common weight regularization techniques like $L_1$ or $L_2$. Even when assuming that each hidden unit in the first layer modeled one interaction, interaction detection was still very accurate. These results lead to the question of whether neural networks automatically separate out interactions at all hidden layers as like Figure 2b when common regularization techniques are applied.

To test this hypothesis, we train 10 trials of ReLU-based feedforward networks of size 4-100-100-100-100-1 on dataset $\mathcal{D}$ with $N = 3\mathrm{e}4$ at equal train/validation/test splits and different regularizations,

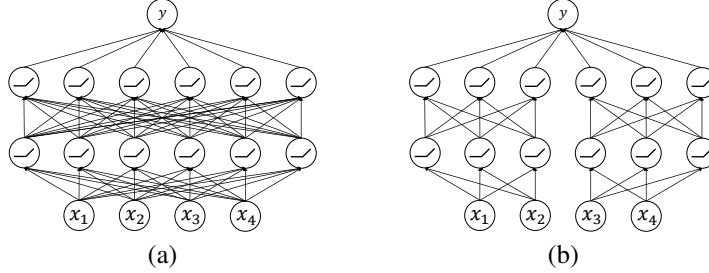

Figure 2: An illustrative comparison between two simple feedforward networks trained on data with interactions $\{1, 2\}$ and $\{3, 4\}$. (a) A standard feedforward neural network, and (b) a desirable network architecture that separates the two interactions. All hidden neurons have ReLU activation, and $y$ is a linear neuron which can precede a sigmoid link function if classification is desired.

which were tuned on the validation set. We then calculate the percentage of hidden units entangling interactions in the first and second hidden layers at different lower thresholds on the magnitudes of all weights (i.e. weight magnitudes below the threshold are zeroed) (see Figure 1). We note that when calculating percentages, if a hidden unit does not learn any of the true interactions, even a superset of one, then that hidden unit is ignored. Consistent with [31], the first hidden layer with $L_1$ or $L_2$ regularization is capable of keeping the interactions separated, but at the second hidden layer, nearly all hidden units entangle the interactions at every threshold when at least one of the interactions is modeled. Therefore, the regularization does very little to prevent the pairwise interactions from entangling within the neural network.

Our understanding that feedforward networks entangle interactions even in the simple setting with two multiplicative interactions motivates the need to disentangle them.

## 4 Disentangling Interactions for Interpretability

In this section, we explain our architecture and regularization framework, NIT, to disentangle interactions in a feedforward neural network.

### 4.1 Architecture Choice

In order to disentangle interactions, we first propose a feedforward network modification that has a dense input weight matrix followed by multiple network blocks, as depicted in Figure 3. Our choice of a dense input weight matrix follows the recent research that a sparse regularized input weight matrix tends to automatically separate out different true feature interactions to different first layer hidden units [31]. Separate block networks at upper layers are used to force the representation learning of separate interactions to be disentangled of each other. The remaining challenge is to ensure that each block only learns one interaction or univariate variable to generate the desired GAM structure (Eq. 1) [1]. Note that in this model, the number of blocks, $B$, must be pre-specified. Two approaches to selecting $B$ are either choosing it to be large and letting sparse regularization cancel unused blocks, or setting $B$ to be small in case a small number of blocks is desired for human interpretation.

In the experiments section (§5.5), we additionally discuss our results on an approach that does not require any pre-specification of network blocks. Instead, this approach attempts to separate interactions through every layer during training rather than rely on the forced separation of blocks.

### 4.2 Disentangling Regularization

As mentioned in the previous section (§4.1), each network block must learn one interaction or univariate variable. Formally, each input hidden unit $h \in \{h_i\}_{i=1}^{p_1/B}$ to block $b \in \{b_i\}_{i=1}^{B}$ must learn the same interaction $\mathcal{I} \in \{\mathcal{I}_i\}_{i=1}^{R}$ or univariate variable $u \in \{u_i\}_{i=1}^{S}$ where $R + S \leq B$. We propose to learn such a model by way of regularization that explicitly defines maximum allowable interaction

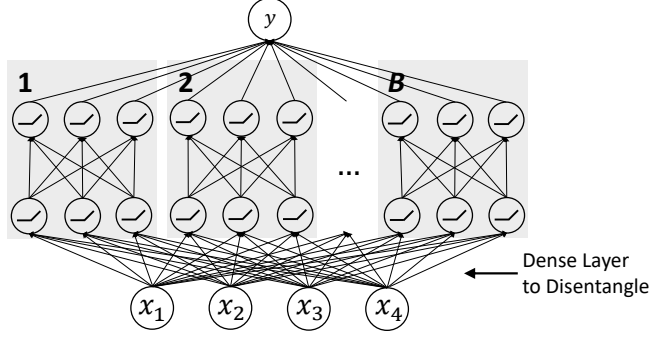

Figure 3: A version of our `NIT` model architecture. Here, `NIT` consists of $B$ mutli-layer network blocks above a common input dense layer. Appropriate regularization on the dense layer forces each block to model a single interaction or univariate variable. This model can equivalently be seen as a standard feedforward neural network with block diagonal weight matrices at intermediate layers.

orders. Fixing the maximum order to be 2 has been the standard in the construction of GAMs with pairwise interactions [20].

The existence of interactions or univariate variables being modeled by first layer hidden units is determined by the nonzero weights entering those units [31]. Therefore, we would like to penalize the number of nonzero elements in rows of the input weight matrix, as a group belonging to a block. The number of nonzero elements can be obtained by using $L_0$ regularization, however, it is known that $L_0$ is non-differentiable and cannot be used in gradient-based regularization [21]. Recently, Louizos, et al. [21] developed a differentiable surrogate to the $L_0$ norm by smoothing the expected $L_0$ and using approximately binary gates $\mathbf{g} \in \mathbb{R}^n$ to determine which parameters $\boldsymbol{\theta} \in \mathbb{R}^n$ to set to zero. The parameters $\boldsymbol{\theta}$ of a hypothesis $\hat{h}$ are then re-parameterized as separate parameters $\tilde{\boldsymbol{\theta}}$ and $\boldsymbol{\phi}$, such that for a dataset of $N$ samples $\{(\mathbf{x}_1, y_1), \dots, (\mathbf{x}_N, y_N)\}$, Empirical Risk Minimization becomes:

$$\mathcal{R}(\tilde{\boldsymbol{\theta}}, \boldsymbol{\phi}) = \frac{1}{N}\left(\sum_{i=1}^{N}\mathcal{L}\left(\hat{h}(\mathbf{x}_i; \tilde{\boldsymbol{\theta}} \odot \mathbf{g}(\boldsymbol{\phi}), y_i\right)\right) + \lambda \sum_{j=1}^{n} g_j(\phi_j), \tag{2}$$

which follows the ideal reparameterization of $\boldsymbol{\theta}$ into $\tilde{\boldsymbol{\theta}}$ and $\mathbf{g}$ as:

$$\theta_j = \tilde{\theta}_j g_j, \quad g_j \in \{0, 1\}, \quad \tilde{\theta}_j \neq 0, \quad \|\boldsymbol{\theta}\|_0 = \sum_{j=1}^{n} g_j \tag{3}$$

We note that Eq. 2 is not in its final form for clarity and in fact uses distributions for the gates to enable differentiability and exact zeros in parameters; we refer interested readers to [21]. We propose a disentangled group regularizer denoted by $\mathcal{L}_{\mathcal{K}}$, to disentangle feature interactions in the `NIT` framework. $\mathcal{L}_{\mathcal{K}}$ is designed to be a group version of the smoothed $L_0$ regularization. Let $\mathbf{G}, \boldsymbol{\Phi} \in \mathbb{R}^{B \times p}$ be matrix versions of the vectors $\mathbf{g}$ and $\boldsymbol{\phi}$ from Eq. 2. Let $\mathcal{T} : \mathbb{R}^{B \times p} \to \mathbb{R}^{p_1 \times p}$ assign the same gate to all first layer hidden units in a block corresponding to a single feature, for all such *groups* of hidden units in every block. Just as $\tilde{\theta}_j \neq 0$ in Eq. 3, $\tilde{W}_{ij}^{(1)} \neq 0, \forall i = 1, \dots, p_1$ and $\forall j = 1, \dots, p$. Then, the cost function for our `NIT` model $f(\cdot)$ has the form:

$$\mathcal{R}_{\texttt{NIT}} = \frac{1}{N}\left(\sum_{i=1}^{N}\mathcal{L}\left(f(\mathbf{x}_i; \tilde{\mathbf{W}}^{(1)} \odot \mathcal{T}(\mathbf{G}(\boldsymbol{\Phi})), \{\mathbf{W}^{(\ell)}\}_{\ell=2}^{L+1}, \{\mathbf{b}^{(\ell)}\}_{\ell=1}^{L+1}, y_i\right)\right) + \mathcal{L}_{\mathcal{K}}, \tag{4}$$

where $\mathcal{L}_{\mathcal{K}}$ is our proposed disentangling regularizer. Since $\mathbf{G}$ is $\approx 1$ when a feature is active in a block and $\approx 0$ otherwise, the estimated interaction order of a block is defined as $\hat{k}_i = \sum_{j=1}^{p} G_{ij}(\Phi_{ij}) \quad \forall i = 1, 2, \dots, B$. Let $\tilde{B} = \sum_{i=1}^{B} \mathbb{1}(\hat{k}_i \neq 0)$. Then, we can conveniently learn generalized additive models with desirable properties by including two terms in our regularizer:

$$\mathcal{L}_{\mathcal{K}} = \underbrace{\max\left\{\left(\max_i \hat{k}_i\right) - K, 0\right\}}_{\text{limits the maximum interaction order to be } K} + \underbrace{\lambda \frac{1}{\tilde{B}} \sum_{i=1}^{B} \hat{k}_i}_{\substack{\text{encourages smaller interaction} \\ \text{orders and block sparsity}}} \tag{5}$$

Table 1: A comparison of the number of models needed to construct various forms of GAMs. $GA^KM$ all interactions refers to constructing a GAM for every interaction of order $\leq K$ respectively. $MLP_{cutoff}$ is an additive model of Multilayer Perceptrons, and $\eta$ is the top number of interactions based on a learned cutoff.

| Framework | GAM original [10] | $GA^KM$ all interactions | $GA^2M$ [20] | $MLP_{cutoff}$ [31] | NIT (proposed) |
|---|---|---|---|---|---|
| # models | $O(p)$ | $O(p^K)$ | $O(p^2)$ | $O(\eta)$ | $O(1)$ |

The first term is responsible for penalizing the maximum interaction order during training to be a pre-specified positive integer $K$. A threshold at $K$ is enforced by a rectifier [8], which is used for its differentiability and sharp on/off switching behavior at $K$. The second term both penalizes the average non-zero interaction order over all blocks and sparsifies unused blocks.

### 4.3 Advantages of our NIT Framework

Constructing generalized additive models by disentangling interactions has important advantages, the main of which is that our NIT framework can learn the GAM in $O(1)$ models, whereas previous methods either needed to learn a model for each interaction, or in the case of traditional univariate GAMs [10], learn a model for each feature (Table 1). Our approach can construct the GAM quickly because it leverages gradient based optimization to determine interaction separations.

In addition to efficiency, our approach is the first to investigate setting hard maximum thresholds on interaction order, for any order $K$. This is a straightforward result of our regularizer formulation.

Previous methods [20, 2] have focused on advocating tree-based GAMs to interpret and visualize what the model learns, and there exist few works which have explored neural network based GAMs with interactions for interpretability. While the tree-based GAMs can provide interpretability, their visualization can appear jagged since decision trees divide their feature space into axis-parallel rectangles which may not be user-friendly [20]. This kind of jagged visualization is not a problem for neural network-based GAMs, which as a result can produce smoother and more intuitive visualizations.

## 5 Experiments

### 5.1 Experimental Setup

We validate the efficacy of our method first on synthetic data and then on four real-world datasets.

Our experiment on synthetic data is the baseline disentangling experiment discussed in §3. We use real-world datasets to evaluate the predictive performance of NIT (under restriction of maximum interaction order $K$) as compared to standard and relevant machine learning models: Linear/Logistic Regression (LR), GAM [19], $GA^2M$ [20], Random Forests (RF), and the Multilayer Perceptron (MLP). We consider standard LR and GAM as the models that do not learn interactions, and $GA^2M$ learns up to pairwise interactions, and RF and MLP are fully-complexity models [20] that can learn all interactions.

After the performance evaluation, we visualize and examine interactions learned by our NIT from a medical dataset. The real-world datasets (Table 2) include two regression datasets previously studied in statistical interaction research: Cal Housing [24] and Bike Sharing [6], and two binary classification datasets MIMIC-III [15] and CIFAR-10 binary. CIFAR-10 binary is a binary classification dataset (derived from CIFAR-10 [17]) with two randomly selected classes, which are "cat" and "deer" in our experiments. We study binary classification as opposed to multi-class classification to minimize learned interaction orders and accord with previous research on GAMs [20]. Root-mean squared error (RMSE) and Area under ROC (AUC) are used as the evaluation metrics for the regression and classification tasks.

Table 2: Real-world datasets

| Dataset | $N$ | $p$ | %Pos |
|---|---|---|---|
| Cal Housing | 20640 | 8 | - |
| Bike Sharing | 17379 | 15 | - |
| MIMIC-III | 20922 | 40 | 14.02% |
| CIFAR-10 binary | 12000 | 3072 | 50.0% |

In all of our NIT models, we set $B = 20$ and use an equal number of hidden units per network block for any given hidden layer. The hyperparameter $\lambda$ in our disentangling regularizer (Eq. 5) was found

Table 3: Predictive performance of `NIT`. RMSE is calculated from standard scaled outcome variables. *GA$^2$M took several days to train and did not converge. Lower RMSE and higher AUC means better model performance.

| Model | Cal Housing | | Bike Sharing | | MIMIC-III | | CIFAR-10 binary | |
|---|---|---|---|---|---|---|---|---|
| | $K$ | RMSE | $K$ | RMSE | $K$ | AUC | $K$ | AUC |
| LR | | $0.60 \pm 0.016$ | | $0.78 \pm 0.021$ | | $0.70 \pm 0.013$ | | $0.676 \pm 0.0072$ |
| GAM | | $0.506 \pm 0.0078$ | | $0.55 \pm 0.016$ | | $0.75 \pm 0.015$ | | $0.829 \pm 0.0014$ |
| GA$^2$M | | $0.435 \pm 0.0077$ | | $0.307 \pm 0.0080$ | | $0.73 \pm 0.012$ | | * |
| `NIT` | 2 | $0.448 \pm 0.0080$ | 2 | $0.31 \pm 0.013$ | 2 | $0.76 \pm 0.011$ | 10 | $0.849 \pm 0.0049$ |
| | 3 | $0.437 \pm 0.0077$ | 3 | $0.26 \pm 0.015$ | 4 | $0.76 \pm 0.013$ | 15 | $0.858 \pm 0.0020$ |
| | 4 | $0.43 \pm 0.013$ | 4 | $0.240 \pm 0.0097$ | 6 | $0.77 \pm 0.011$ | 20 | $0.860 \pm 0.0034$ |
| RF | | $0.435 \pm 0.0095$ | | $0.243 \pm 0.0053$ | | $0.685 \pm 0.0087$ | | $0.793 \pm 0.0034$ |
| MLP | | $0.445 \pm 0.0081$ | | $0.22 \pm 0.012$ | | $0.771 \pm 0.0096$ | | $0.860 \pm 0.0046$ |

by running a grid search for validation performance on each fold of a 5-fold cross-validation [2]. In our experiments, we don't assume $K$, so we report the performances of `NIT` when varying $K$. Learning rate was fixed at $5e-2$ while the disentangling regularization was applied. For the hyperparameters of baselines and those not specific to `NIT`, tuning was done on the validation set. For all experiments with neural nets, we use the ADAM optimizer [16] and early stopping on validation sets.

## 5.2  Training the `NIT` Framework

Model training in `NIT` was conducted in two phases. The first was a disentangling phase where each block learned one interaction or univariate variable. The second phase kept $\mathcal{L}_{\mathcal{K}} = 0$ and $\mathbf{G}$ fixed, so that $\mathbf{G}$ acted as a mask that deactivated multiple features in each block. The second phase of training starts when the maximum interaction order across all blocks [3] was $\leq K$. and the maximum interaction order of the disentangling phase stabilizes. We also reinitialized parameters between training phases in case optimization was stuck at local minima.

## 5.3  Disentangling Experiment

We revisit the same function $x_1 x_2 + x_3 x_4$ that the MLP failed at disentangling (§3) and evaluate `NIT` instead. We train 10 trials of `NIT` with a 4-100-100-100-100-1 architecture like before (§3) and a grid search over $K$. In Figure 1 we show that `NIT` disentangles the $x_1 x_2$ and $x_3 x_4$ pairwise interactions at all possible lower weight thresholds while maintaining a performance (RMSE = $1.3e-3$) similar to that of MLP. Note that the architecture choice of `NIT` (Figure 3) automatically disentangles interactions in the entire model when the first two hidden layers are disentangled.

## 5.4  Real-World Dataset Experiments

On real-world datasets (Table 2), we evaluate the predictive performance of `NIT` at different levels of $K$, as shown in Table 3. For the Cal Housing, Bike Sharing, and MIMIC-III datasets, we choose $K$ to be 2 first and increase it until `NIT`'s predictive performance is similar to that of RF or MLP. For CIFAR-10 binary, we set $K = 10, 15, 20$ to demonstrate the capability of `NIT` to learn high-order interactions. The exact statistics of learned interaction orders for all datasets are shown in appendix A in the supplementary materials. For all the datasets, the predictive performance of `NIT` is either comparable to MLP at low $K$, or comparable to GA$^2$M at $K = 2$ and RF/MLP at higher values of $K$, as expected.

In Figure 4, we provide all visualizations of what `NIT` learns at $K = 2$ on one fold of the MIMIC-III dataset. MIMIC-III is currently the largest public health records dataset [15], and our prediction task is classifying whether a patient will be re-admitted into an intensive care unit within 30 days. Since $K = 2$, all learned interactions are plotted as heatmaps as shown, and the remaining univariate variables are shown in the left six plots. We notice interesting patterns, for example when a patient's

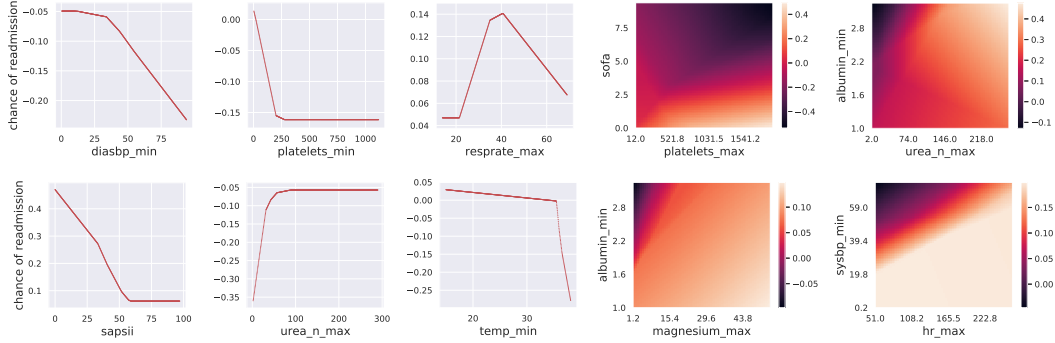

Figure 4: Visualizations that provide a global [27] and transparent [30] interpretation of `NIT` trained on the MIMIC-III dataset at $K = 2$. Outcome scores are interpreted as contribution to 30-day hospital readmission in the same way described by [2]. The output bias of `NIT` is 0.21.

minimum temperature rises to $\approx 40°$C, the chance for readmission drops sharply. Another interesting pattern is in an interaction plot showing as sofa score (which estimates mortality risk) increases, the chance for readmission decreases. We checked that these potentially un-intuitive patterns are indeed consistent with those in the actual dataset by examining the frequency of readmission labels relative to temperature or sofa score. This insight may warrant further investigation by medical experts.

## 5.5 Disentangling Interactions Through All Layers

In addition to disentangling at the first weight matrix, we discuss results on modifying `NIT` to disentangle interactions through all layers' weight matrices. By doing this, we no longer require $B$ network blocks nor group $L_0$. Instead, $L_0$ is applied to each individual weight as done normally in [21]. We now define layer-wise gate matrices $\mathbf{G}^{(1)}, \mathbf{G}^{(2)}, \ldots, \mathbf{G}^{(L)}$ of gates for each weight in the corresponding matrices $\mathbf{W}^{(1)}, \mathbf{W}^{(2)}, \ldots, \mathbf{W}^{(L)}$. The estimated interaction order $\hat{k}_i$ from Eq. 5 is now for each neuron $i$ in the last hidden layer and $\hat{k}_i = \sum_{j=1}^{p} [\sigma(\mathbf{G}^{(L)} \mathbf{G}^{(L-1)} \ldots \mathbf{G}^{(1)})]_{ij}$, where normalized matrix multiplications between $\mathbf{G}^{(\ell)}$'s are taken. Here, $\sigma$ is a sigmoid-type function, $\sigma(\mathbf{G}') = \frac{\mathbf{G}'}{c+|\mathbf{G}'|}$, which approximates a function that sends all elements of $\mathbf{G}'$ greater than or equal to 1 to be 1, otherwise 0 ($c$ is a hyperparameter satisfying $0 < c \ll 1$). Theoretical justification for the formulation of $\hat{k}_i$ is provided in Appendix E.

Disentangling interactions through all layers can perform well in regression tasks. When we let this approach discover the max interaction order by setting $K = 0$ for $\mathcal{L}_{\mathcal{K}}$ in Eq. 5 and $c = 1\mathrm{e} - 2$, `NIT` is able to reach 0.43 RMSE at max order 3 for Cal Housing, and 0.26 RMSE at max order 6 for Bike Sharing without re-initializing models. Now without network blocks, `NIT` architectures are smaller than before (Appendix D), i.e. 8-200-200-1 for Cal Housing and 15-300-200-100-1 for Bike Sharing.

## 5.6 Limitations

Although `NIT` can learn a GAM with interactions in $O(1)$ models, the disentangling phase in our training optimization can take longer with increasing $p$ or $B$. In addition, if $K$ is not pre-specified, a search for an optimal $K$ for peak predictive performance can be a slow process when testing each $K$. Finally, since optimization is non-convex, there is no guarantee that correct interactions are learned.

## 6 Conclusion

We have investigated an interpretation of a feedforward neural network based on its entangling of interactions, and we proposed our framework, `NIT`, for disentangling the interactions to learn a more interpretable neural network. To disentangle interactions, we developed a way of maintaining separations in the network and applying a direct penalty on interaction orders. `NIT` corresponds to reducing a feedforward neural network to a generalized additive model (GAM) with interactions, allowing it to learn the GAM in $O(1)$ models. In experiments, we have demonstrated the effectiveness of `NIT` at obtaining high predictive performance at different $K$ and the value of `NIT` for visualization. For future work, we would like to study ways of making `NIT` perform at state-of-the-art levels for interaction detection in addition to obtaining high predictive performance under $K$-order constraints.

**Acknowledgments**

We thank Umang Gupta and anonymous reviewers for their generous feedback. This work was supported by National Science Foundation awards IIS-1254206, IIS-1539608 and a Samsung GRO grant.

## Footnotes

[1]It is possible that multiple blocks learn the same interaction or univariate variable, and some blocks can learn a subset of what other blocks learn. In such cases, the duplicate or redundant blocks can be combined. We leave the prevention of this type of learning for future work.

[2]For all real-world datasets except CIFAR-10 binary, 5-fold cross-validation was used, where model training was done on 3 folds, validation on the 4th fold and testing on the 5th. For the CIFAR dataset, the standard test set was only used for testing, and an inner 5-fold cross-validation was done on an $80\%$-$20\%$ train-validation split.

[3]With this criteria, $K \neq p$, but the criteria can be changed to allow $K = p$.

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
