[Supplementary Material · supplementary_camera_ready.pdf]

## Supplementary Materials

## A   Learned Interaction Orders

Shown in Table 4 are statistics of learned interaction orders by `NIT` corresponding to experiments in Table 3. Interaction orders at different $K$ are averaged over 5 folds of cross-validation. The interaction orders of the baselines are the maximum interaction orders they can learn.

Table 4: Interaction order statistics when all repeated interactions and sparsified blocks are ignored. An order of 1 is a univariate variable.

| Model | | Cal Housing | | Bike Sharing | | MIMIC-III | | CIFAR-10 binary | |
|---|---|---|---|---|---|---|---|---|---|
| | | $K$ | order | $K$ | order | $K$ | order | $K$ | order |
| LR/GAM | | - | 1 | - | 1 | - | 1 | - | 1 |
| GA$^2$M | | - | 2 | - | 2 | - | 2 | - | 2 |
| `NIT` | max | 2 | $2.0 \pm 0.0$ | 2 | $2.0 \pm 0.0$ | 2 | $2.0 \pm 0.0$ | 10 | $10 \pm 0.0$ |
| | mean | | $1.70 \pm 0.057$ | | $1.83 \pm 0.067$ | | $1.5 \pm 0.12$ | | $5.8 \pm 0.20$ |
| | min | | $1.0 \pm 0.0$ | | $1.0 \pm 0.0$ | | $1.0 \pm 0.0$ | | $1.8 \pm 0.75$ |
| | max | 3 | $3.0 \pm 0.0$ | 3 | $3.0 \pm 0.0$ | 4 | $4.0 \pm 0.0$ | 15 | $13.6 \pm 0.49$ |
| | mean | | $2.2 \pm 0.27$ | | $2.69 \pm 0.054$ | | $3.0 \pm 0.24$ | | $9 \pm 1.1$ |
| | min | | $1.2 \pm 0.40$ | | $1.6 \pm 0.49$ | | $1.2 \pm 0.40$ | | $4 \pm 1.9$ |
| | max | 4 | $3.8 \pm 0.40$ | 4 | $4.0 \pm 0.0$ | 6 | $5.6 \pm 0.80$ | 20 | $19 \pm 1.2$ |
| | mean | | $2.8 \pm 0.37$ | | $3.6 \pm 0.11$ | | $4.1 \pm 0.80$ | | $12.7 \pm 0.95$ |
| | min | | $1.2 \pm 0.40$ | | $1.8 \pm 0.75$ | | $1.8 \pm 0.75$ | | $4 \pm 2.4$ |
| RF/MLP | | - | 8 | - | 15 | - | 40 | - | 3072 |

## B   Importance of G

Shown in Table 5 is a comparison between the original prediction performance of `NIT` and its performance when the learned gates **G** within each network block are shuffled before the second phase of training (§5.2). The lowered predictive performance due to shuffling is more pronounced for lower interaction orders and different datasets. For example, large performance differences are observed for the Bike Sharing dataset. The smaller performance differences with the MIMIC-III and CIFAR-10 binary datasets may be due to high feature correlations.

Table 5: The sensitivity of predictive performance to shuffling of the learned gates **G** within each network block.

| Model | Cal Housing | | Bike Sharing | | MIMIC-III | | CIFAR-10 binary | |
|---|---|---|---|---|---|---|---|---|
| | $K$ | RMSE | $K$ | RMSE | $K$ | AUC | $K$ | AUC |
| original | 2 | $0.448 \pm 0.0080$ | 2 | $0.31 \pm 0.013$ | 2 | $0.76 \pm 0.011$ | 10 | $0.849 \pm 0.0049$ |
| shuffled | | $0.51 \pm 0.089$ | | $0.6 \pm 0.13$ | | $0.73 \pm 0.019$ | | $0.841 \pm 0.0047$ |
| original | 3 | $0.437 \pm 0.0077$ | 3 | $0.26 \pm 0.015$ | 4 | $0.76 \pm 0.013$ | 15 | $0.858 \pm 0.0020$ |
| shuffled | | $0.47 \pm 0.013$ | | $0.5 \pm 0.15$ | | $0.74 \pm 0.022$ | | $0.854 \pm 0.0036$ |
| original | 4 | $0.43 \pm 0.013$ | 4 | $0.240 \pm 0.0097$ | 6 | $0.77 \pm 0.011$ | 20 | $0.860 \pm 0.0034$ |
| shuffled | | $0.440 \pm 0.0082$ | | $0.33 \pm 0.042$ | | $0.75 \pm 0.029$ | | $0.860 \pm 0.0027$ |

## C   Performance Without Re-initializing `NIT`

In §5.2, disentangling proceeds until the desired interaction order is reached for a patience duration, after which disentangling ends and parameters are re-initialized to complete training. Without re-initializing, the performance of `NIT` at $K = 2$ on Cal-Housing was $0.475$ RMSE on average, which is worse than our $0.448$ score. Likewise, other pre-reinitialization scores at the lowest reported $K$s were also worse: Bike-Sharing $0.330$, MIMIC-III $0.731$, and CIFAR-10 binary $0.746$.

## D  Hyperparameters

Hyperparameters of NIT not mentioned in the paper body but used in our main experiments (Table 3) are shown in Table 6. Architecture sizes are from input through each hidden layer to output. $\lambda_2$ is the $L_2$ regularization constant we used on the second phase of training NIT after disentangling (§5.2).

Table 6: Hyperparameters of NIT omitted from the main paper corresponding to experiments in Table 3

| hyperparameter | Cal Housing | Bike Sharing | MIMIC-III | CIFAR-10 binary |
|---|---|---|---|---|
| architecture | 8-400-300-200-100-1 | 15-800-600-400-200-1 | 40-200-100-1 | 3072-400-300-200-100-1 |
| $\lambda_2$ | $1e-5$ | $1e-5$ | $1e-4$ | $1e-5$ |

The architectures of the MLP baselines vary based on hyperparameter tuning and are shown in Table 7. As before, the architecture sizes are from the input through each hidden layer to output. All baseline MLPs are tuned with $L_2$ regularization.

Table 7: Archiectures of the MLP baselines

| Cal Housing | Bike Sharing | MIMIC-III | CIFAR-10 binary |
|---|---|---|---|
| 8-140-100-60-20-1 | 15-100-100-100-1 | 40-300-200-100-1 | 3072-200-200-1 |

Finally, in our approach of disentangling interactions through all layers (§5.5), we set $\lambda = 5e-3$ for both Cal Housing and Bike Sharing. We also use a regularization constant in front of the max term in Eq. 5, which is $0.5$ for Cal Housing and $0.05$ for Bike Sharing. As before, the learning rate is set to $5e-2$. All other hyperparameters were already mentioned in §5.5 and §5.1.

## E  Interaction Orders from Multiplying Layer-wise Gate Matrices

Below, we provide theoretical support for matrix multiplying gate matrices $\mathbf{G}^{(\ell)}$ of each layer $\ell = 1, 2, \ldots, L$ in our estimation of interaction orders, to disentangle interactions through all layers.

**Lemma 2** (Paths From Multiplying Gate Matrices). *Let $f(\cdot)$ be a feedforward neural network. Assume that in general, weights in $\mathbf{W}^{(\ell)}$ are nonzero $\forall \ell = 1, \ldots, L+1$. Let $\mathbf{G}^{(1)}, \mathbf{G}^{(2)}, \ldots, \mathbf{G}^{(L)}$ be masks for corresponding weight matrices $\mathbf{W}^{(1)}, \mathbf{W}^{(2)}, \ldots, \mathbf{W}^{(L)}$, where the elements of each mask are binary $\{0, 1\}$. Let $\tilde{\mathbf{G}} \in \mathbb{R}^{p_L \times p}$ be given by the matrix multiplications $\mathbf{G}^{(L)} \mathbf{G}^{(L-1)} \ldots \mathbf{G}^{(1)}$. Then a nonzero value of $\tilde{G}_{ij}$ indicates that there is a nonzero weighted path from feature $j$ to neuron $i$ in the $L$-th hidden layer, and a zero value of $\tilde{G}_{ij}$ indicates there is no such path.*

*Proof.* In the case that $f(\cdot)$ has a single hidden layer ($L = 1$), $\tilde{\mathbf{G}} = \mathbf{G}^{(1)}$ directly gives the zero and nonzero paths from features to the $L$-th hidden layer.

In cases where $f(\cdot)$ has more than one hidden layer, first consider the weight connectivity between input features and the second hidden layer. Since a feedforward neural network is a directed acyclic graph where a hop transitions from one layer to the next, we can view the connectivity from input features to the second hidden layer as two hops or two applications of an adjacency matrix, $\mathbf{A}$, comprising of $\mathbf{G}^{(1)}$ and $\mathbf{G}^{(2)}$ as:

$$
\mathbf{A} = \left(
\begin{array}{c|c|c}
\mathbf{0} & \left(\mathbf{G}^{(1)}\right)^{\top} & \mathbf{0} \\
\hline
\mathbf{0} & \mathbf{0} & \left(\mathbf{G}^{(2)}\right)^{\top} \\
\hline
\mathbf{0} & \mathbf{0} & \mathbf{0}
\end{array}
\right)
$$

Therefore, the adjacency matrix for two hops is:

$$\mathbf{A}^2 = \left( \begin{array}{cc|c} \mathbf{0} & \mathbf{0} & \left( \mathbf{G}^{(2)} \mathbf{G}^{(1)} \right)^{\top} \\ \hline \mathbf{0} & \mathbf{0} & \mathbf{0} \\ \hline \mathbf{0} & \mathbf{0} & \mathbf{0} \end{array} \right)$$

Since the elements of $\mathbf{A}^2$ are the number of paths between graph vertices in two hops, the nonzero elements of $\mathbf{G}^{(2)} \mathbf{G}^{(1)}$ represent the existence of paths from features to the second hidden layer, and the zero elements represent the lack of such paths. We can therefore repeatedly add hops up to the $L$-th hidden layer, yielding $\mathbf{G}^{(L)} \mathbf{G}^{(L-1)} \ldots \mathbf{G}^{(1)}$ to represent the zero and nonzero paths from features to the neurons in the $L$-th layer.

$\square$

It follows that the feature interaction order at neuron $i$ in hidden layer $L$ can be calculated as $\hat{k}_i = \sum_{j=1}^{p} \sigma(\tilde{\mathbf{G}})_{ij}$, where in this case $\sigma(\cdot)$ serves the purpose of setting all nonzero elements in $\tilde{\mathbf{G}}$ to one. We note that the matrix multiplications in calculating $\tilde{\mathbf{G}}$ can cause numeric overflow; therefore, we recommend normalizing the result of each matrix multiplication, i.e. dividing the result by its maximum magnitude.

## F    Runtime Considerations

The runtime of NIT depends on the size of networks and the number of blocks, $B$, used during training. In our main experiments (§5.4) where $B$ is always 20, runtime advantages are primarily seen when $p$ is large, such as the case for MIMIC-III ($p = 40$) and CIFAR-10 binary ($p = 3072$) experiments. For example, the average runtime of NIT across $K$ values in Table 3 is 1.5 hours for MIMIC-III (versus 45 hours using GA$^2$M) and 55 minutes for CIFAR-10 binary (versus 10 hours using GAM without interactions). In the cases where $p$ is small such as Cal Housing and Bike Sharing, the training time caused by large networks (Appendix D) and the task itself can dominate. In particular for regression tasks, we recommend using the variant of NIT that disentangles interactions through all layers (§5.5) to speed up training as this approach does not require large networks that arise from $B$ blocks.