[Reviews · NeurIPS 2018]

Reviewer 1



This paper proposes a novel approach to more interpretable learning in neural networks. In particular, it addresses the common criticism that the computations performed by neural networks are often hard to intuitively interpret, which can be a problem in applications, e.g. in the medical or financial fields. The authors suggest adding a novel regulariser to the weights of first layer of a neural network to discover and preserve non-additive interactions in the data features up to a chosen order and preserve these relationships without entangling them together (e.g. x1*x2 + x3*x4 would result in hidden neurons representing the interactions {x1, x2} and {x3, x4} but not {x1, x2, x3, x4}). These interactions could then be further processed by the separate columns of the neural network. The approach was evaluated on a number of datasets and seems to perform similarly to the baselines on regression and classification tasks, while being more interpretable and less computationally expensive. While this work appears promising, I find that a few details are missing that would make me fully convinced about the utility of this approach. I am willing to increase my score if these questions are adequately addressed. Suggestions for improvement: -- Section 2.1. It might be worth expanding this section to give the reader a bit more intuition about the problem set up. I found it hard to understand the notation when first reading the paper, since I was lacking the high level picture of what the authors were trying to achieve. The notation became clearer once I finished reading the paper. -- Line 122. Are there any non-linearities in your network? -- Line 123. It would be useful to know how the data was generated. In particular, what ranges were the x_{1-4} sampled from? How were the train/validation/test splits done? Currently it is hard to assess whether there was a danger for significant overlap in the splits due to the small ranges/inadequate splits. -- Figure 1. This is an important motivational figure, yet it is hard to understand due to the lack of detail. For example, % neurons entangled metric is explained in a single line in a footnote. This is something that require a lot more details to understand. Furthermore, the plot is small and the majority of lines are impossible to see. -- Line 202. What is the architecture for your MLP baseline? How does it compare to the architecture of your approach. What optimiser was used for the experiments? -- Section 5.1. Why do you only use binary classification datasets? It would be interesting to see how the model performs on the full MNIST dataset. -- Line 228. "phase stabilises" - what does that mean? "Reinitialised parameters" - would be good to see the results when the parameters are not reinitialised. -- Line 252. How did you check whether these patterns really exist in the dataset? Minor details: -- Line 48. The acronym GAM is used for the first time without explaining what it means. It should be introduced in line 34. -- Line 62. The authors are encouraged to include a reference to the beta-VAE work (Higgins et al, 2017) -- GAMs section (lines 65-74). This section might read better before the Interpretability section. -- Line 83. I think there should be a comma after [p] ("...all input features [p], with [I]>=2..."), as otherwise the definition reads as being circular -- Line 203. "fully-complexity models" - not sure what that means -- Line 258. optimisation in non-convex --> optimisation is non-convex ___________________________________________________ After reading the authors' response, I am happy to increase my score to 6. I hope the authors increase the readability of their paper for the camera ready version as promised in their response.

Reviewer 2



The paper proposes a variation of feedforward neural networks that does not learn (potentially unwanted) non-additive interactions between variables above a certain order. The variation includes a block-based architecture with a sparsifying layer and a regularizer that caps the interaction order within blocks. If the max interaction order is <= 2, then all interactions can be easily visualized as independent plots. The authors draw an analogy between this model and GAMs with interactions. Pros: - The paper is relevant to NIPS and reasonably well written. - Combining blocking and soft-capping the interaction order is an interesting idea and a natural extension to GAMs. - Some results are nice: the proposed model can reach the same performance but better visualizability than the competitors. Cons: - The definition of interactions in this paper is not equivalent to the one given in [28], which affects both the main message and the results. It is also confusing. - Evaluation is lacking, see below. Major remarks: - The paper focuses on non-additive interactions. However, definition 1 captures additive interactions as well. Consider for instance the function f(x) = x1 + x2 + x3 + x4, which has no non-additive interactions at all. Also let all variables be non-negative. One way to write f(x) as a feedforward is: f(x) = 1*h1(x) + 1*h2(x) + 0 h1(x) = max{1*x1 + 1*x2 + 0, 0} h2(x) = max{1*x3 + 1*x4 + 0, 0} Although this network has no non-additive interactions at all, definition 1 would classify all variables as entangled. In other words, the presence of a path of positive weights does not imply that there is a non-additive interaction. That is, the definition is incorrect. Indeed, the authors do not show that it is equivalent to the actual definition of non-additive interactions given in [28]. Now, both the regularizer and the results depend on this definition. This implies that: - the regularizer controls a *surrogate* (actually, an upper bound) of the # of non-additive interactions. - the # of interactions reported in the results may overestimate the real # of non-additive interactions. To summarize, unless I'm missing something, the method does not do what the authors claim it does, and the same goes for the experiments. The appropriate claims should be toned down accordingly. - In order to see if DI really works, two things should be checked: (i) the first sparsifying layer works, (ii) the cap on maximum interaction order works. Point (ii) has been validated empirically with real-world datasets. The same is not true for point (i). The synthetic dataset is extremely simple (it is sampled from an order 2 homogeneous polynomial). This is not enough to show that the sparsifying layer, especially when coupled with the cap on the interaction order, works as intended. It should be easy to show that it does by testing it on *several* of order K polynomials (with varying K), and checking that the first layer learns the intended independencies. The experiments with realistic datasets do not help in this case, because the gold standard is unknown. - I also have a few other (neither major nor minor) questions about the experiments: - The output layer of the proposed method is linear. Is this also the case for binary classification tasks? - I am a bit puzzled that there is no comparison to kernel methods (where the interaction order can be capped very easily, e.g. with polynomial kernels). Why were they left out of the comparison? - It seems that training these models may be possibly complicated, potentially hindering reproducibility. Will the code/data be made available? - What about run-time comparisons? Minor remarks: - Figure 1 is very confusing. I suggest to use colors to identify different models/regularizers and linestyles for the different layers. The legend should also be sorted accordingly. - l 110: a function the in the form - l 152: to learn such form of model -> to learn such a model - l 194: "since neural networks are known to learn smooth functions". This may be the case in general, but it is definitely *not* the case for the ReLU networks used in this paper. It is well known that ReLU networks with linear outputs learn piecewise linear functions (see [Exact and Consistent Interpretation for Piecewise Linear Neural Networks: A Closed Form Solution, Chu et al., 2018] for a nice visualization), which are definitely not smooth. The learned function may rensamble a smooth function, but this is only the case if the target function itself is smooth. So the claim that the proposed method encourages smooth plots should be either removed or qualified appropriately. - l 217: set to a small value [and] then increased - l 216: "if K was not given ... model". This is very confusing. Please just say that you report the performance of models while varying K. (Otherwise it sounds like a single value of K was chosen unfairly.) ==== After reading the rebuttal, I decided to increase my score. The issues I found can be either solved by rewording the paper (namely clarifying that the method uses a surrogate upper-bound of the # of non-additive independencies), or are not too major (they concern "only" the first layer).

Reviewer 3



This is an interesting paper that proposes a novel method for using a neural network to efficiently learn a GAM model with interactions by applying regularization to disentangle interactions within the network. They show that the resulting model achieves similar performance to an unrestricted neural network model on the chosen datasets. I would want to see this approach tested on more typical neural network application domains, such as image classification (e.g. CIFAR-10). The only image data they used was MNIST, which is much simpler than most image datasets. The chosen data sets have a relatively small number of features, so it's not clear if the approach would yield similar results on richer data and thus provide a general way to make neural networks more interpretable. Richer data could also have a high optimal value of K, which would further slow down the search process. It would also be useful to give more detail about the MLP model to show that it is a strong baseline. Does it have the same number of units and layers as the DI model? Section 4.3 states that "a neural-network based GAM like ours tends to provide a smoother visualization" - it is unclear to me what 'smoothness' means here. If I understood correctly, the visualizations in Figure 4 were produced by plotting variables and interactions that were found by the learned GAM model, and I don't see how the visualizations produced by a tree-based GAM model would be different. It would help to clarify this in the paper. The paper is generally well-written and clear. As a minor nitpick, in Figure 4, it would be helpful to label the y-axis and legend in the plots as "chance of readmission". The caption refers to [2] for an exact description of the outcome scores - adding outcome labels to the plots would make the figure more self-contained. UPDATE after reading the author feedback: The rebuttal has addressed my concerns, so I have increased my score. It was particularly good to see the authors were able to provide results for CIFAR-10. I am still a bit concerned that the MLP baselines are smaller than the corresponding DI models, so it might not be an entirely fair comparison.